# Recommendations in pre-registrations and internal review board proposals promote formal power analyses but do not increase sample size

**Marjan Bakker**[1]*, **Coosje L. S. Veldkamp**[2], **Olmo R. van den Akker**[1], **Marcel A. L. M. van Assen**[1,3], **Elise Crompvoets**[1,4], **How Hwee Ong**[5], **Jelte M. Wicherts**[1]

1 Department of Methodology and Statistics, Tilburg University, Tilburg, The Netherlands, 2 Faculty of Social Sciences, Utrecht University, Utrecht, The Netherlands, 3 Department of Sociology, Utrecht University, Utrecht, The Netherlands, 4 Cito National Institute for Educational Measurement, Arnhem, The Netherlands, 5 Department of Social Psychology, Tilburg University, Tilburg, The Netherlands

* M.Bakker_1@uvt.nl

**Data Availability Statement:** The data and materials for this study are available at https://osf.io/94f2w/.

## Abstract

In this preregistered study, we investigated whether the statistical power of a study is higher when researchers are asked to make a formal power analysis before collecting data. We compared the sample size descriptions from two sources: (i) a sample of pre-registrations created according to the guidelines for the Center for Open Science Preregistration Challenge (PCRs) and a sample of institutional review board (IRB) proposals from Tilburg School of Behavior and Social Sciences, which both include a recommendation to do a formal power analysis, and (ii) a sample of pre-registrations created according to the guidelines for Open Science Framework Standard Pre-Data Collection Registrations (SPRs) in which no guidance on sample size planning is given. We found that PCRs and IRBs (72%) more often included sample size decisions based on power analyses than the SPRs (45%). However, this did not result in larger planned sample sizes. The determined sample size of the PCRs and IRB proposals (Md = 90.50) was not higher than the determined sample size of the SPRs (Md = 126.00; $W$ = 3389.5, $p$ = 0.936). Typically, power analyses in the registrations were conducted with G*power, assuming a medium effect size, α = .05 and a power of .80. Only 20% of the power analyses contained enough information to fully reproduce the results and only 62% of these power analyses pertained to the main hypothesis test in the pre-registration. Therefore, we see ample room for improvements in the quality of the registrations and we offer several recommendations to do so.

## Introduction

Many studies in the psychological literature are underpowered [1–4]. Specifically, in light of the typical effect sizes and sample sizes seen in the literature, statistical power of psychological studies is estimated to be around .50 [1] or even .35 [2, 3]. This means that the probability of

**Funding:** JW recieved a Vidi grant (no. 452-11-004) from the Netherlands Organization for Scientific Research (NWO; www.nwo.nl) and a Consolidator Grant (IMPROVE) from the European Research Council (ERC; grant no. 726361; https://erc.europa.eu/). The funders had no role in study design, data collection and analysis, decision to publish, or preparation of the manuscript.

**Competing interests:** The authors have declared that no competing interests exist. Jelte M. Wicherts is a PLOS ONE Editorial Board member. This does not alter the authors' adherence to PLOS ONE Editorial policies and criteria.

making a Type II error (failing to reject the null hypothesis when it is false) of a typical study is between 50% and 65%. At the same time, the psychological literature shows a very high prevalence of positive outcomes (estimates range from 91 to 97%; [5–7]). This means that many studies end up in a file drawer or that researchers use some researcher degrees of freedom opportunistically to get a positive (i.e., statistically significant) result during the analysis or reporting of results. The opportunistic use of these researchers degrees of freedom (also called Questionable Research Practices, QRPs) seems to be widespread [8–15] and is seen as one of the reasons that many psychological studies fail to replicate [16, 17]. When studies are well-powered, researchers are less inclined to use researchers degrees of freedom opportunistically because the true effects show up more easily and the influence of the opportunistic use of researchers degrees of freedom on the study results is less severe (e.g., removing or keeping an outlier has less impact on the statistical result when the sample size is large; [2]). Thus, running well-powered studies will eventually result in a more reliable scientific literature filled with fewer biased outcomes.

A possible solution to increase the statistical power of psychological studies is to carry out and report a power analysis a priori, instead of basing sample size on the common practice in a field, or some general rule of thumb [18]. The power of statistical tests depends on the nominal significance level (typically .05), the sample size, and the effect size in the population, such as Cohen's $d$ for between-group mean comparisons. Since the significance level and the population effect size are often fixed, the general way to increase the statistical power of a study is to increase the sample size. A power analysis is therefore used to determine the sample size needed to reach the intended power level given the estimated effect size and significance level. Several programs are currently available to conduct a power analysis, such as G*Power [19] and the pwr package for R [20]. The most difficult part of a power analysis is to get a good estimation of the population effect size, since this value is unknown and often no good (meta-analytic) estimation is available.

The low statistical power of psychological studies and the importance of a priori power analyses have been known for a long time [21], but despite a longstanding debate, statistical power in psychological studies remains low [1, 2, 4, 22]. Furthermore, in a recent study, only half of psychology researchers indicated that they typically used a power analysis to make sample size decisions [22]. Other common ways to determine sample size were practical constraints in for example the available time or money, some rule of thumb (e.g., 20 subjects in each cell), or simply following the common practice in a given field. The study by Bakker et al. [22] also showed that researchers highly overestimate the power of studies when effect sizes are small to medium-sized, which are quite common in psychology [2, 3, 23]. Because of all these underpowered studies, and the major consequences of this, the general advice is to do an a priori power analysis to determine the sample size needed to find the estimated effect.

Maddock and Rossi [18] showed that federally funded studies, which require that a power analysis is completed before a grant is submitted and will generally also have the resources to test a larger sample, had significantly higher power to detect medium and small effects than studies that did not receive extramural funding. Two other ways to motivate researchers to conduct a formal power analysis before starting an experiment are to incorporate the power analysis in a pre-registration template or in an Institutional Review Board (IRB, also sometimes called Ethics Review Board) proposal form. Therefore, in the current study, we want to investigate whether the statistical power of a study is higher when researchers are asked to conduct a power analysis before collecting the data as part of a pre-registration or an IRB proposal.

Pre-registration entails the specification of hypotheses, study design, and data analysis plans for a study prior to data collection. The idea is that pre-registration prevents that choices

during data-analysis (e.g., which outcome variable, what to do with outliers or missing data) are consciously or unconsciously influenced by the outcome of the analysis (e.g., whether the outcome is significant or not) [9, 24–31]. In the present research, we evaluated two pre-regis-tration templates that are available on the Open Science Framework (OSF): a simple template that only asks the researcher to describe the study and thereby offering maximal flexibility to the researcher to define pre-registration content that is most fitting for their research ("Stan-dard Pre-Data Collection"; SPR), and the most extensive format that provides a specific work-flow and instructions for what must be preregistered ("Prereg Challenge"; PCR). The Prereg Challenge format includes recommendations about doing a power analysis to decide on the intended sample size, whereas the Standard Pre-Data Collection format does not.

Another way to motivate researchers to do a formal power analysis before conducting a study is by making it part of an IRB proposal. The primary goal of an IRB is to protect the rights and welfare of people who participate in research. Researchers are therefore asked to write an IRB proposal in which they describe the possible risks and harms of the research to the participants. This proposal is then reviewed by the board before the study is conducted. Although a power analysis to determine the sample size is not typically part of an IRB proposal, it can be argued that a study with very low statistical power is not informative at all and thereby a waste of the valuable time of the research participants. Therefore, the IRB from Tilburg School of Behavioral and Social Sciences includes a question in which researchers need to sub-stantiate their intended sample size, preferably using a power analysis.

In this study, we wanted to investigate whether asking researchers explicitly to do a power analysis in a pre-registration or IRB proposal results in more well powered studies. We used these three different registration types (SPR, PCR, and IRB) to answer the following research questions: (1) Does making formal sample size decisions before data collection as part of a pre-registration or an IRB proposal lead to more power-based sample size decisions than in pre-registrations in which this is not explicitly asked? (2) Does making formal sample size deci-sions before data collection as part of a pre-registration or an IRB proposal result in larger sample sizes than in pre-registrations in which this is not explicitly asked? Additionally, we examine the power analyses themselves by answering (3) What are the typical values of the dif-ferent input parameters of the power analyses? (4) Are those power analyses correctly calcu-lated? And, (5) what are the differences in typical input values between power analyses as part of a pre-registration and power analyses as part of IRB proposals? Based on the answers to these research questions, we will offer some concrete recommendations to improve power analyses and increase the statistical power of scientific studies.

The first two research questions resulted in the following three preregistered hypotheses:

1. Pre-registrations created according to the guidelines for the Center for Open Science Pre-registration Challenge (PCRs) and IRB proposals from Tilburg School of Behavior and Social Sciences (IRBs) lead to more power-based sample size decisions than pre-registra-tions created according to the guidelines for Open Science Framework Standard Pre-Data Collection Registrations (SPRs). More specifically, we expected that the proportion yes on Q2a (specified in the methods section) would be higher for the PCRs and IRBs than for the SPRs.

2. PCRs and IRBs result in higher intended sample sizes than SPRs. More specifically, we expected the average score on Q11/Q15 (specified in the methods section) to be higher for the PCRs and IRBs than for the SPRs.

3. Doing a power analysis mediates the effect of registration type (PCRs and IRBs vs. SPRs) on the intended sample size as investigated in Hypothesis 2. Specifically, we expected, besides a

significant second hypothesis, an effect of doing a power analysis (Q2a) on the intended sample size (Q11/Q15) after controlling for registration type.

No hypotheses were specified for the latter three research questions. The complete pre-registration of our study can be found at https://osf.io/pgw5q/ and deviations from our pre-registration are discussed at the end of the methods section.

## Method

### Sample

**Selection of pre-registrations.**   Data for the current project were collected together with the data for another project that evaluated the extent to which current pre-registrations restricted the opportunistic use of 29 researcher degrees of freedom [32]. At the start of our study, on August 17, 2016, 5,829 publicly available pre-registrations were listed on the pre-registrations search page on the OSF (https://osf.io/search/?q=*&filter=registration&page=1). These registrations included all types of pre-registrations then available on the OSF. The Center for Open Science provided URLs to the 122 public PCRs that had been submitted as of August 16, 2016. Following our pre-registration, we randomly selected 53 PCRs and 53 SPRs. See Bakker et al. [32] for more details on the selection of pre-registrations.

**Selection of IRB proposals.**   We considered IRB proposals that were submitted to the ethical board of TSB between January 1, 2010, and November 15, 2017. An IRB contact person approached the authors of these IRB proposals and asked them whether they would allow the IRB to share only the part of the proposal about their sample size decision with us. When researchers agreed, only those parts of the IRB proposals were selected and anonymized and then shared with us (this procedure was approved by the Ethics Review Board of Tilburg School of Social and Behavioral Sciences: EC-2017.Ex83). Of the 380 IRB proposals from 143 different researchers, we got permission to include 189 IRB proposals (50%). After inspection of the IRB proposals, we found that ten proposals contained multiple studies since researchers can also submit an IRB proposal for a research line. We decided to evaluate these studies separately. This resulted in 25 additional studies. Furthermore, we encountered three duplicate studies and one missing study (submitted to another ethical board). Further inspection of the 210 descriptions of studies in the IRB proposals showed that 55 contained sample size descriptions for qualitative studies. Even though we did not preregister this, we excluded these from our sample because power analyses are not applicable to these studies. This resulted in 155 included IRB proposals.

### The scoring protocol

To compare the different registration types on how the sample size was determined and on how the power analysis was performed, we developed a scoring protocol (https://osf.io/ue9gy/) with the following items: Q1) whether the word power was mentioned (yes/no), and Q2) how the authors decided on the sample size. For this second question multiple categories could be selected (e.g., both a power analysis and some practical constraint could be mentioned). The categories were: power analysis (Q2a), practical constraints (Q2b), rule of thumb (Q2c), common practice in the field (Q2d), or as many participants as possible (Q2e). These categories correspond to those in the survey by Bakker et al. [22].

If a power analysis was used to determine sample size, we scored the reported power analysis. If multiple power analyses were reported, we scored the power analysis that resulted in the determined sample size. If none of the reported power analyses resulted exactly in the determined sample size, we scored the power analysis that resulted in the sample size smaller than

and closest to the intended sample size. For the scoring of the power analysis we used the following items: Q3) number of power analyses, Q4) software program that was used, Q5) effect size measure, Q6) effect size value, Q7) rationale for effect size estimate (Cohen's values, earlier studies, literature, pilot studies, only interested in effect size values larger than *x*, other, or not specified), Q8) α, Q9) sidedness of the test (one-sided, two-sided, unspecified, Q10) the level of power that they wanted to reach, and Q11) determined sample size. We selected Cohen's values for the effect size rationale (Q7) if they referred to a small, medium, or large effect size and used the threshold values as specified in Cohen [33], which are also used as threshold values in G*Power.

Furthermore, we checked Q12) whether enough information was provided to replicate the power analysis (yes/no). If enough information was provided, we also checked Q13) whether the reported power analysis matched the hypothesis under investigation (yes/no) and Q14) whether we got the same results when we reproduced the power analysis (yes/no). If no power analysis was used to decide on the sample size we collected only Q15) the determined sample size.

## Coding procedure

Each SPR, PCR, and IRB proposal was independently coded by two of the six experienced coders. As part of the study by Bakker et al. [32], the coders first scored the SPRs and PCRs on the extent to which the registrations restricted the potential opportunistic use of researchers degrees of freedom. Subsequently, the coders gleaned the power analyses from the proposals using the scoring protocol (https://osf.io/ue9gy/) and reported their results in a coding sheet (https://osf.io/p3sfd/). When finishing the coding, the coders' scores were compared with an R script (https://osf.io/yq2z4/) and any differences were resolved by discussion. Across all items, registration types and coding pairs, the exact same answer had been given in 66% of the cases. However, many of the discrepancies consisted of differences in the exact wording or differences in the use of capital letters. Coders were always able to come to an agreement, so no third coder was needed.

## Statistical analyses

To test our first hypothesis that PCRs and IRB proposals lead to more power based sample size decisions than SPRs, we used a $\chi^2$ test to compare the proportion of PCRs and IRB proposals together with the proportion of SPRs that used a power analysis to make a sample size decision (Q2a = yes). To test our second hypothesis that PCRs and IRB proposals lead to larger sample sizes than SPRs, we applied the Mann-Whitney-Wilcoxon (MWW) test to the intended sample sizes (Q11 and Q15) of the different registrations. The MWW test was used because this non-parametric test takes possible violations of assumptions and the presence of outliers into account, and has the best power and Type I error rate control when the distribution is skewed or when outliers are present [34]. To test our third hypothesis that doing a power analysis mediates the effect of registration type on the intended sample size, we used a joint significance test. The joint significant test is both powerful [35] and easy to apply when the indirect effect is the product of two different types of statistical effects, as in our case where one effect is binary (doing a power analysis or not) and the other is continuous (i.e., sample size).

For the joint significance test, we ran a logistic regression of the effect of registration type on doing a power analysis (a) and a linear regression with sample size as dependent variable and registration type (c') and doing a power analysis (b) as predictors. If the effect of doing a power analysis is also significant (after controlling for registration type), we obtain joint significance as evidence for a mediation effect. To estimate this mediation effect, we calculated the

difference between c and c' by doing an additional linear regression analysis to estimate the effect of c (type of registration as predictor of sample size). We compared c with c', the effect of registration type on sample size, after controlling for doing a power analysis. We have three tests for our hypotheses, which are all directional. Therefore, we only used one-sided tests for our three tests and used a Bonferroni correction to correct for the three tests, resulting in α = .0167 (one-sided).

Because this project was a joint project with Bakker et al. [32], the sample size (SPR and PCR) was based on their main analysis (Wilcoxon-Mann-Whitney U test to test the difference in average restriction scores between the two types of pre-registrations) and the IRB proposals are added to that. Bakker et al. required a total sample size of 106 (53 per group), which is, therefore, our minimum sample size. Our power analysis indicated that this minimum sample size would be able to detect a medium effect (w = .3 for our first hypothesis; d = .5 for our second hypothesis) with .80 power when α = .05 (one-tailed). For the mediation effect (third hypothesis), we used the joint significance test which would need 77 participants when both a and b are medium effects [36]. See our pre-registration (https://osf.io/djx5b/) for a full description of our power analysis.

To answer our other research questions (all explorative), we planned to present the descriptives on all the scoring items for the three registration types separately. To investigate the differences between the registration types, we used Fisher's exact tests to compare proportions and robust ANOVAs with 20% trimmed means to compare numerical values. To control the family-wise error rate of our explorative analyses, we used a Holm correction. For our analyses, we used an R script (https://osf.io/ec36u/; R version 3.0.1) that was more extensive than our preregistered script (https://osf.io/djx5b/) because it also included the explorative analyses.

### Deviations from our pre-registration

We started the coding phase of our study with 53 SPRs and 53 PCRs, but during the coding phase one of the PCRs turned out to have been withdrawn by the authors of the pre-registration. As the coding phase could not be finalized for this pre-registration, we excluded this pre-registration from our data file. Our final sample thus consisted of 53 SPRs and 52 PCRs. Furthermore, because of a delay in our study, we decided to approach the researchers of all IRB proposals up and to November 15, 2017, instead of only including those submitted before August 31, 2016, as we pre-registered. Lastly, 55 IRB proposals contained qualitative studies, which we excluded from our sample because power analyses are not applicable to these studies.

## Results

### Confirmatory hypotheses

Our final sample consisted of 53 SPRs, 52 PCRs, and 155 IRB proposals. Our first hypothesis was supported; of the 207 PCRs and IRB proposals 150 (72%) made a power based sample size decision, whereas 24 (45%) of the 53 SPRs made a power based sample size decision ($\chi^2(1)$ = 14.083, $p < .001$, $\varphi$ = .233). Our second hypothesis was not supported; the determined sample size of the PCRs and IRB proposals (Median = 90.50, Mean = 212.41, SD = 395.86, range = [6 – 3200]) was not larger than the determined sample size of the SPRs (Median = 126.00, Mean = 221.58, SD = 246.48, range = [6 – 1200]; $W$ = 3389.5, $p$ = 0.936, $d$ = -0.025). Because the registration type did not predict sample size (second hypothesis, and effect c), we could not proceed with the joint significance mediation test of our mediation (third) hypothesis.

For completeness, we also present the results of the two regression analyses that were preregistered to test our third hypothesis. The logistic regression on the effect of registration type

on doing a power analysis shows a significant effect of registration type (b = 0.715, $p < .001$; note that this is essentially the same as the test of our first hypothesis). The linear regression with sample size as dependent variable and registration type and doing a power analysis (Q2a) as predictors, shows a significant effect of power analysis in the opposite direction (b = -213.07, $p < .001$) and no significant effect of registration type (b = 20.67, $p = .745$).

## Other descriptive and explorative results

In Table 1 we present the descriptives (proportions, medians) of all the different scoring items for the SPRs, PCRs and IRB proposals separately to answer our third and fourth research question (i.e., what are the typical values of the different input parameters of the power analyses and are the power analyses correctly calculated). In this Table, we also present the Holm corrected $p$ values and effect sizes of the tests that compared the different registration types to answer our fifth research question (i.e., explore the differences between the three registration types).

**Deciding on sample size.**   Power was most often mentioned in the IRB proposals (83% versus 51% and 60% for SPRs and PCRs, respectively), and a power based sample size decision was most often made in IRB proposals (79%), compared to SPRs (45%) and PCRs (54%). Sample size decisions were also based on practical constraints (mentioned by 15% overall, with the highest percentage, 38%, for PCRs), some rule of thumb (18% overall, highest, 25%, for IRB proposals), or other comparable studies (10% overall, highest, 25%, for PCRs). Two percent of the registrations mentioned that they wanted to use as many participants as possible.

**Typical power analysis.**   To answer our third and fifth research questions we investigated the typical values of the different input parameters of the power analyses and checked for differences between the different registration types on these input parameters. Only one power analysis was reported in 148 of the 174 (85%) SPRs, PCRs, and IRB proposals that used a power analysis to decide on sample size. Twenty proposed studies (11%) reported two power analyses, while six (3%) reported three or more power analyses. In 72 (41%) power analyses authors failed to mention which software program was used. Ninety-six (55%) of the power analysis were carried out using G*power and six (3%) used another program. These other programs were Daniel Soper's statistical power calculator (4 times), the pwr R package (once), and the GLIMMPSE calculator (once). We did not find significant differences between the SPRs, PCRs, and the IRB proposals in the number of reported power analyses per planned study or the software program used.

In 52 (30%) of the reported power analyses in the SPRs, PCRs, and IRB proposals the effect size type (e.g., Cohen's $d$) was not specified and in 26 (15%) the value of the effect size was not specified. The most commonly used effect size types were $d$, $r$, $f$, and $f^2$, which were reported 38, 15, 34, and 12 times, respectively). The mean, median, and range for these effect sizes are given in Table 2. Other effect size measures that were used were $d_z$, Hedges' $g$, β, SD, $w$, (partial) $\eta^2$, $R^2$, Odds Ratio, and a proportion change. All effect sizes for which this was possible are transformed into Cohen's $d$ (all except β, SD, $w$, partial $\eta^2$, and proportion change). The mean, median, and range of the transformed effect sizes are given in the bottom row of Table 2. When we compared the (transformed) Cohen's $d$ of the three different registration types with a robust ANOVA, we found no statistically significant difference ($F(2, 21.57) = 0.228$, $p = 1.000$). Medium or a slightly below medium effect size was most commonly specified. This corresponds with the results of an earlier survey in which researchers estimated the ES of a typical psychological study to be $d = 0.39$ [22]. Seventy-eight (45%) of the SPRs, PCRs, and IRB proposals failed to specify where the effect size estimate was based on. This happened most often in the IRB proposals (50%) and least often in the PCRs (18%). If specified, the ES

**Table 1. Descriptives of all the items in the protocol for each of the registration types separately.**

| | SPR | PCR | IRB | Total | ES[a] | p[b] |
|---|---|---|---|---|---|---|
| N | 53 | 52 | 155 | 210 | | |
| Mention power (Q1) | 27 (51%) | 31 (60%) | 128 (83%) | 186 (72%) | 0.304 | < .001 |
| Sample size based on: | | | | | | |
| Power (Q2a) | 24 (45%) | 28 (54%) | 122 (79%) | 174 (67%) | 0.310 | < .001 |
| Practical constraints (Q2b) | 6 (11%) | 20 (38%) | 13 (8%) | 39 (15%) | 0.330 | < .001 |
| Rule of Thumb (Q2c) | 4 (8%) | 5 (10%) | 38 (25%) | 47 (18%) | 0.204 | .082 |
| Other studies (Q2d) | 2 (4%) | 13 (25%) | 12 (8%) | 27 (10%) | 0.245 | .021 |
| As many participants as possible (Q2e) | 0 (0%) | 1 (2%) | 4 (3%) | 5 (2%) | 0.073 | 1.000 |
| N. of power analyses (Q3) | | | | | 0.174 | .874 |
| 1 | 16 (67%) | 26 (93%) | 106 (87%) | 148 (85%) | | |
| 2 | 5 (21%) | 2 (7%) | 13 (11%) | 20 (11%) | | |
| 3 or more | 3 (13%) | 0 (0%) | 3 (2%) | 6 (3%) | | |
| Program (Q4) | | | | | 0.125 | 1.000 |
| G*power | 11 (46%) | 12 (43%) | 73 (60%) | 96 (55%) | | |
| Other | 0 (0%) | 1 (4%) | 5 (4%) | 6 (3%) | | |
| Not specified | 13 (54%) | 15 (54%) | 44 (36%) | 72 (41%) | | |
| ES type (Q5) not specified | 4 (17%) | 8 (29%) | 40 (33%) | 52 (30%) | 0.120 | 1.000 |
| ES value (Q6) not specified | 2 (8%) | 5 (18%) | 19 (16%) | 26 (15%) | 0.078 | 1.000 |
| ES based on: (Q7) | | | | | 0.315 | .028 |
| Cohen's values | 5 (21%) | 6 (21%) | 25 (20%) | 36 (21%) | | |
| Earlier study | 5 (21%) | 12 (43%) | 17 (14%) | 34 (20%) | | |
| Literature | 1 (4%) | 3 (11%) | 16 (13%) | 20 (11%) | | |
| Pilot study | 0 (0%) | 2 (7%) | 0 (0%) | 2 (1%) | | |
| Only interested in large ES | 1 (4%) | 0 (0%) | 0 (0%) | 1 (1%) | | |
| Other | 0 (0%) | 0 (0%) | 3 (2%) | 3 (2%) | | |
| Not specified | 12 (50%) | 5 (18%) | 61 (50%) | 78 (45%) | | |
| α (Q8) | | | | | 0.100 | 1.000 |
| .05 | 15 (63%) | 21 (75%) | 77 (63%) | 113 (65%) | | |
| Other value | 1 (4%) | 0 (0%) | 10 (8%) | 11 (6%) | | |
| Not specified | 8 (33%) | 7 (25%) | 35 (29%) | 50 (29%) | | |
| Sidedness of the test (Q9) | | | | | 0.164 | 1.000 |
| One-sided | 5 (21%) | 2 (7%) | 5 (4%) | 12 (7%) | | |
| Two-sided | 1902 (8%) | 5 (18%) | 19 (16%) | 26 (15%) | | |
| Not specified | 17 (71%) | 21 (75%) | 98 (80%) | 136 (78%) | | |
| Power (Q10) | | | | | 0.125 | 1.000 |
| .8 | 14 (58%) | 17 (61%) | 67 (55%) | 98 (56%) | | |
| Other | 9 (38%) | 9 (32%) | 32 (26%) | 50 (29%) | | |
| Not specified | 1 (4%) | 2 (7%) | 23 (19%) | 26 (15%) | | |
| Sample size[c] | | | | | | |
| Median | 126 | 90 | 92 | 99.5 | | 1.000[d] |
| Not specified | 13 (25%) | 2 (4%) | 5 (3%) | 20 (8%) | 0.320 | < .001 |
| Complete (Q12) | 9 (38%) | 4 (14%) | 21 (17%) | 34 (20%) | 0.183 | 1.000 |
| Relevant (Q13) | 5 (21%) | 3 (11%) | - | 8 (15%) | 0.086 | 1.000 |
| Correct (Q14) | 8 (33%) | 4 (14%) | 21 (17%) | 33 (19%) | 0.149 | 1.000 |

[a] Cramer's V for all fisher exact tests

[b] Holm corrected p values

[c] based on all included registrations and IRB proposals

[d] This is a robust ANOVA with 20% trimmed means.

**Table 2. Mean (M), Median (Md), [range], and frequency (N) of the different effect sizes used in the power analyses (Cohen's d, r, f, and f², effect sizes transformed to Cohen's d) for each of the registration type separately.**

| Effect size | SPR | PCR | IRB |
|---|---|---|---|
| Cohen's d | M = 0.55; Md = 0.50; [0.27–1.41]; N = 10 | M = 0.54; Md = 0.50; [0.30–1.00]; N = 9 | M = 0.48; Md = 0.50; [0.20–1.00]; N = 19 |
| r | M = 0.26; Md = 0.26; [–]; N = 1 | M = 0.25; Md = 0.23; [0.11–0.43]; N = 4 | M = 0.26; Md = 0.25; [0.015–0.60]; N = 10 |
| f | M = 0.20; Md = 0.23; [0.10–0.25]; N = 4 | M = 0.13; Md = 0.14; [0.10–0.15]; N = 3 | M = 0.25; Md = 0.25; [0.10–0.60]; N = 27 |
| f² | M = 0.35; Md = 0.35; [–]; N = 1 | - | M = 0.09; Md = 0.10; [0.02–0.15]; N = 11 |
| *Transformed to Cohen's d* | M = 0.52; Md = 0.50; [0.20–1.41]; N = 19 | M = 0.49; Md = 0.42; [0.20–1.00]; N = 17 | M = 0.52; Md = 0.50; [0.03–1.50]; N = 78 |

According to Cohen (33) the threshold values are for Cohen's d 0.2, 0.5, and 0.8, for r 0.1, 0.3, and 0.5, for f 0.1, 0.25, and 0.4, and for f² 0.02, 0.15, and 0.35 for small, medium, and large effect sizes, respectively. These are also the threshold values as used in G*power.

estimate was most often based on Cohen's values (21%), a specific earlier study (20%) or the literature in general (11%). Only one ES estimate was set to a high value because the researchers were only interested in a large effect, two (1%) were based on a pilot study, and three (2%) on some other reason (e.g., based on norm scores or theory).

The α was not specified in 50 (29%) of the power analyses. Of the remaining 124 power analyses, 113 (91%) reported using α = .05. The 9% that reported another α reported an α smaller than .05 (6 times) or probably involved typos or other errors (e.g., .5 or .95; reported 5 times). In twelve (7%) of the power analyses it was specified that a one-sided test was used in the power analysis, and in 26 (15%) it was reported that a two-sided test was used. The sidedness is not relevant for the power analysis of several tests (e.g., tests based on the F distribution). However, in 39 (65%) of the 60 power analyses that were based on a test for which the sidedness of the test is required information (Cohen's $d$, $d_z$, $r$, and Hedges' $g$) the sidedness was not specified. Intended power was not specified in 26 (15%) of the power analyses. In the remaining 148 power analyses, 98 (66%) used a power of .8, while other common values were .9 (11 times; 7%) and .95 (28 times; 19%).

The intended sample size was not reported in 13 (25%) of the SPRs, compared to only 2 (4%) and 5 (3%) for the PCRs and IRB proposals, respectively. Of all 174 reported power analyses, only two (1%) failed to mention the intended sample size (both in IRB proposals). When no power analysis was reported, 18 (21%) failed to mention the intended sample size. The three registration types did differ in this regard ($p$ = .021; $V$ = 0.296), with the highest number of missing intended sample sizes (13; 45%) for the SPRs. In our confirmatory analyses, we already considered intended sample sizes and found no significant effect of registration type (SPRs versus the PCRs and IRB proposals together). We also used JASP [37] to do a Bayesian Mann-Whitney-Wilcoxon test to estimate the evidence for the null hypothesis (no difference between the two registration types). This test showed substantial evidence for the null hypothesis of no difference ($BF_{01}$ = 5.280). When we compare the intended sample size of the three different registration types with a robust ANOVA, we also found no statistically significant difference ($F(2, 46.47)$ = 1.151, $p$ = 1.000). In the pre-specified analyses for our third hypothesis, we found that using a power analysis to decide on sample size was related to lower intended sample size (see note 3). However, the assumption of linearity was not met since the distribution of the intended sample sizes was skewed. A robust one-way ANOVA that takes this violation into account did not show a significant difference between PCRs and IRB proposals together and SPRs ($F(1, 45.36)$ = 2.829, $p$ = 1.000). We also investigated a potential interaction effect between registration type and doing a power analysis on the intended sample size, but this interaction was not significant either ($F$ = 2.277, $p$ = 1.000).

**Reproducibility and correctness of power analyses.** To answer our fourth research question, we first checked whether we had enough information to reproduce the reported power analyses. To run an accurate power analysis one would require at least the effect size type and value, the α level, and the intended power level. As shown in Table 1 and discussed above, this information was often lacking. Furthermore, an exact reproduction of a power analysis requires additional information, like the sidedness of the test or the correlation between repeated measures. Power analyses were scored as completely reported (Q12) if all information required by G*power was given. Only 34 (20%) power analyses provided enough information to reproduce the power analysis.

For the completely reported power analyses in the SPRs and PCRs, we checked whether the reported power analysis matched the hypothesis under investigation (Q13). Note, that we could not do this for the IRB proposals, because we had no access to the full proposals. Of the 13 completely reported power analyses, 8 (62%) matched the hypothesis. For the completely reported power analyses, we also checked whether we obtained the same results when we reproduced the power analysis. We arrived at the same intended sample size in 33 (97%) of the 34 power analyses with complete information.

## Discussion

In our preregistered study we investigated whether more power analyses are reported when the guidelines of a pre-registration or IRB proposal recommends doing a power analysis and whether power analyses were associated with larger intended sample sizes. We found support for our first hypothesis and found that the PCRs and IRB proposals reported more power based sample size decisions than the SPRs. The number of power based sample size decisions in both the PCRs and IRB proposals (72%) was also much higher than the percentage of researchers who indicated to do a power analysis as their typical way to decide on sample size (47%; [22]). Our first recommendation is, therefore, to make power analyses part of a pre-registration template or IRB proposal guidelines, because it increases the number of reported formal a priori power analyses.

However, our results clearly showed that making power analyses part of a pre-registration template or IRB proposal guidelines is not enough. We expected that reporting power analyses would be associated with larger intended sample sizes, but our results did not find support for the second preregistered hypothesis. PCRs and IRB proposals did not report a higher determined sample size than the SPRs. Therefore, we could not find support either for our preregistered third hypothesis that the planned sample size was higher for PCRs and IRB proposals because they more often conducted a power analysis.

Furthermore, we found in our explorative analyses that power analyses are often reported incompletely or conducted in inappropriate ways. It was striking that only 20% of the power analyses contained enough information to fully reproduce the results. Often the intended level of power (15%), α (29%), effect size type (30%), or effect size value (15%) was missing. If enough information *was* available to reproduce the results, the results were almost always correctly reported (research question 4). However, 38% reported a power analysis for a statistical test that was different from the main statistical test used to test their hypothesis. Our second recommendation, therefore, is that researchers should provide enough details about the power analysis to make power analyses more reproducible. This can be done straightforwardly by copying the results from G*Power (or another program) or sharing the relevant code. Our third recommendation is that researchers first decide on the statistical analyses they will use to test their hypotheses and only then select the appropriate power analysis/analyses.

The most challenging part of a power analysis is to get a good estimate of the population effect size. We found that these estimates are often based on the sample estimate of a single previous study (20%) or the literature in general (11%; this included effect size estimates based on a meta-analysis). Although this might seem appealing, these sample and meta-analytic effect size estimates will often overestimate the population effect size because of uncertainty and publication bias [38, 39]. Furthermore, earlier research showed that intuitions for the exponential form of power functions are often flawed [22]. We can illustrate this with an example in which we want to determine the sample size of a study that investigates the mean difference between two independent groups when $\alpha$ = .05 and power is 0.80. The population effect size might be estimated based on a previous study in which an effect of $d = 0.5$ is found, with a $CI_{95}$ that ranges from $d = 0.3$ to $d = 0.7$. The estimated sample size for the point estimate ($d = 0.5$) is 64 participants in each group, whereas the estimates for the lower and upper bound of the $CI_{95}$ will result in an estimated sample size of 176 and 34 participants in each group, respectively. The lower bound of the $CI_{95}$ of an effect size estimate will often be close to zero in many single small studies, which will result in enormous sample sizes. A meta-analysis will generally give a more precise effect size estimate (e.g., the $CI_{95}$ will be smaller), but this estimate might still be biased. Our fourth recommendation is, therefore, that the meta-analytical effect sizes are first corrected for publication bias [40, 41] and that researchers perform a power analysis for a range of effect size estimates. This may help researchers to understand the relationship between effect size and statistical power better and will let them think more deeply about what effect size they would consider as meaningful [42].

To answer the fifth research question, we investigated the differences between the three registration types in an explorative analysis. We did not find differences between the three registration types on most of the items of our protocol. However, we found the highest percentage of power analysis based sample sizes in the IRB proposals. Thus, the confirming evidence for our first hypothesis might be mainly caused by the relatively high number of power analyses in the IRB proposals. One explanation is that by asking the researchers of TSB to share their IRB proposals with us, mainly the researchers who are confident about their sample size rationale have shared their IRB proposals and thereby biasing the results. Another explanation of the difference between PCRs and IRB proposals in the number of power based sample sizes might be the way these pre-registrations and proposals are reviewed. The review form of the IRB proposal asks specifically whether the proposal contains a substantiated power analysis. The PCRs are also reviewed, but these reviews do not specifically focus on the power of the study. It might therefore that a good review procedure increases the number of power analyses. Another way of implementing a review procedure concerns registered reports [43]. Registered reports are peer-reviewed before collecting the data and are published independently of the final results. In this way, registered reports help to prevent low statistical power, selective reporting of results, and publication bias. It is of course possible that SPRs conducted a power analysis to determine the sample size, but did not report the power analysis, because it was not asked for in the pre-registration format. However, we expect that a power analysis typically will be included when it is conducted while writing the pre-registration. Other differences are that the sample sizes are most often based on practical constraints and on the typical study for PCRs, while a rule of thumb was most common in the IRB proposals. Furthermore, we found a difference between the registrations types in the explanations of where the estimated effect size came from. Most notable was the high proportion of power analyses in PCRs (43%) that used the effect size found in a single earlier study as effect size estimate in the power analysis. Of the power analyses that were carried out, we see that the typical power analysis (research question 3) was done with G*power, used a medium effect size, $\alpha$ = .05, and a power of .80.

A possible explanation of not finding support for our second hypothesis might be that the smaller determined sample sizes were less often reported in the SPRs, because researchers are more inclined to include the positive aspects of their planned study in the pre-registration (e.g., large sample sizes and well powered designs). We did find in our explorative analyses that the intended sample size in SPRs was indeed missing more often compared to the PCRs and IRB proposals. In this study, we only investigated the determined sample sizes that were reported in the registrations and IRB proposals *before* the actual data were collected. Therefore, we have no information on the final sample sizes of these studies. In a follow-up project, we will compare the final publications with the pre-registrations, which will allow us to compare the planned with the final sample sizes and also to get information on the sample sizes that are currently missing. This will also allow us to compare the final sample sizes of the SPRs and PCRs and check whether doing a power analysis increases the final sample sizes and thereby improves the reliability of studies.

The typically low statistical power of studies and its negative consequences have been discussed for a long time [21]. Although we found that more power analyses are reported and that the planned sample sizes were larger than earlier found in psychological studies (cell sizes of 20–24; [2, 44, 45]), most of the incorporated studies are still unlikely to detect a true but smaller than medium effect. Furthermore, the current 'crisis of confidence', [46] as evidenced by the failure to replicate many prominent findings [16, 47], shows clearly that psychology has a long way to go. So besides clear sample size planning, possibly as part of a pre-registration, IRB proposal, or registered report, a more general shift to larger sample sizes in psychological research is still needed. Two initiatives that offer possibilities to increase sample size are Study-Swap (https://osf.io/view/StudySwap/) and the Psychological Science Accelerator [48] in which different research groups work together to increase sample sizes. We hope that these initiatives will finally result in better-powered research in psychology which results we can trust.

## Author Contributions

**Conceptualization:** Marjan Bakker.

**Data curation:** Marjan Bakker.

**Formal analysis:** Marjan Bakker.

**Investigation:** Marjan Bakker, Coosje L. S. Veldkamp, Olmo R. van den Akker, Marcel A. L. M. van Assen, Elise Crompvoets, How Hwee Ong, Jelte M. Wicherts.

**Methodology:** Marjan Bakker, Coosje L. S. Veldkamp, Marcel A. L. M. van Assen, Jelte M. Wicherts.

**Project administration:** Marjan Bakker.

**Resources:** Marjan Bakker, Coosje L. S. Veldkamp, Marcel A. L. M. van Assen.

**Supervision:** Marjan Bakker.

**Writing – original draft:** Marjan Bakker.

**Writing – review & editing:** Marjan Bakker, Coosje L. S. Veldkamp, Olmo R. van den Akker, Marcel A. L. M. van Assen, Elise Crompvoets, How Hwee Ong, Jelte M. Wicherts.

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
