## [Decision Letter · Decision Letter 0]

19 Dec 2019

PONE-D-19-20027

We have the (1-β)! Requirements in preregistrations and IRB proposals promote formal power analyses

PLOS ONE

Dear Dr. Bakker,

Thank you for submitting your manuscript to PLOS ONE. After careful consideration, we feel that it has merit but does not fully meet PLOS ONE’s publication criteria as it currently stands. Therefore, we invite you to submit a revised version of the manuscript that addresses the points raised during the review process.

As you will see, the reviewers differed in their comments about the merits and limitations of your paper, and both expressed a number of important issues. I have read the manuscript very carefully and found it very interesting and highly relevant. However, I agree with the reviewers that the original contribution of the study should be better specified (in the Introduction and in the Conclusion), and the results more deeply explained. Additionally, we recommend to fully revise the manuscript according to reviewers' minor suggestions.

All the comments are enclosed for your information.

We would appreciate receiving your revised manuscript by Feb 02 2020 11:59PM. To enhance the reproducibility of your results, we recommend that if applicable you deposit your laboratory protocols in protocols.io, where a protocol can be assigned its own identifier (DOI) such that it can be cited independently in the future. For instructions see: http://journals.plos.org/plosone/s/submission-guidelines#loc-laboratory-protocols

We look forward to receiving your revised manuscript.

Kind regards,

Mariagrazia Benassi

Academic Editor

PLOS ONE

Journal Requirements:

2. We noticed you have some minor occurrence of overlapping text with the following previous publication, which needs to be addressed: Bakker, Marjan, et al. "Researchers’ intuitions about power in psychological research." Psychological science 27.8 (2016): 10. In your revision ensure you cite all your sources (including your own works), and quote or rephrase any duplicated text outside the methods section. Further consideration is dependent on these concerns being addressed.

Reviewers' comments:

Reviewer's Responses to Questions

**Comments to the Author**

1. Is the manuscript technically sound, and do the data support the conclusions?

Reviewer #1: Yes

Reviewer #2: Yes

2. Has the statistical analysis been performed appropriately and rigorously? 

Reviewer #1: No

Reviewer #2: Yes

3. Have the authors made all data underlying the findings in their manuscript fully available?

Reviewer #1: Yes

Reviewer #2: Yes

4. Is the manuscript presented in an intelligible fashion and written in standard English?

Reviewer #1: Yes

Reviewer #2: Yes

5. Review Comments to the Author

Reviewer #1: In this preregistered study, the authors tested the effects of pre-registration type (OSF Standard Pre-Data Collection, OSF Prereg Challenge, Tilburg School of Behavioral and Social Sciences Institutional Review Board proposal) on researchers’ decisions about the rationale for the determination of sample size (e.g., whether a power analysis is conducted or not) and on the intended sample size. In line with the authors’ hypotheses, the results showed that the use of power analysis was more frequent in studies that were pre-registered according to the guidelines of OSF Prereg Challenge (PCR) form and of Tilburg School of Behavioral and Social Sciences Institutional Review Board (IRB) proposal than in studies that were pre-registered according to the guidelines of OSF Pre-Data Collection (SPR) form. Inconsistently with the authors’ hypotheses, the intended sample size was not affected by pre-registration type. The results also showed that power analyses are often characterized by technical mistakes and/or by incomplete descriptions.

Overall the manuscript is clear and well-written, and it focuses on a highly debated issue such as the reliability of psychological research. I agree with the authors’ concerns and recommendations about the need for well-conducted power analyses to increase the reliability of psychological research. Nevertheless, as regards this specific study, I have some important concerns that preclude my recommendation for publication in Plos One.

MAJOR POINTS

1) The possible impact of the results. It is unclear to me how the current study can contribute to the extant literature on research methods in psychology. The first result, which is in line with the authors’ expectation, is that power analysis is more likely to be performed when researchers are required to do it, but in my view, this just confirms the obvious. The second (and this time unexpected) finding is that pre-registration type does not appear to affect the intended sample size. The meaning of this finding remains obscure, but I suspect that it can be partly explained by the fact that, in SPRs, power analyses might have been conducted but not reported (see Major Point 3). The third result is that power analyses are often conducted in inappropriate ways (e.g., estimating effect size on a single previous study), and/or they are incorrectly reported. In my view, this is the most interesting and potentially useful outcome of this study, as it shows that simply recommending (or forcing) researchers to conduct a power analysis does not guarantee the reliability of psychological research. I believe that this manuscript would have been more interesting if it focused on this latter point.

2) The rationale of the pre-registered hypotheses and interpretation of the results. The authors suggest that both the IRB proposal and the PCR form require the participants to conduct a formal power analysis. However, as far as I could see from the OSF website, the PCR form (https://osf.io/dn3c4/) simply asks the researchers to clearly state the rationale for sample size determination. It appears that arbitrary constraints such as time and money are considered to be adequate justifications. Therefore, the PCR does not put that emphasis on power analysis, as it is instead suggested by the authors. On the contrary, power analysis probably plays a crucial role in the approval of IRB proposals. Therefore, I do not understand why the authors included PCRs and IRB proposals in the same group. These a priori objections are supported by the results, which show that the relative frequency of power analysis was much higher for IRB proposals (79%) than for PCRs (54%). On page 11, the authors affirm: “Our first hypothesis was supported; of the 207 PCRs and IRB proposals 150 (72%) made a power-based sample size decision, whereas 24 (45%) of the 53 SPRs made a power-based sample size decision …”. I believe that this is a highly misleading description of the results, due to the inappropriate averaging of the results for IRB proposals and PCRs. The percentage of PCRs in which a power analysis was performed (54%) was much more similar to that of SPRs (45%) than to that of IRB proposals (79%). I think that these objections cast severe doubts both on the rationale of the pre-registered hypotheses and on the interpretation of the results.

3) Interpretation of the results. I believe that there is some confusion between reporting and conducting a power analysis. For the PCRs and the IRB proposals, it seems reasonable to assume that if a power analysis was not reported then it had not been conducted. However, this assumption seems to be much less reasonable for the SPRs. Indeed, the SPR form does not require the researchers to specify the rationale for the determination of sample size, which leaves open the possibility that, at least in some cases, a power analysis was actually conducted but it was simply not reported. This appears to invalidate the rationale of the pre-registered hypotheses

MINOR POINTS

1) Title: I believe that the use of acronyms should be avoided in the title.

2) Table 1, “Sample size based on…”: the sum of the percentages is much higher than 100%. Am I missing something?

3) Typos:

Line 127: preregistration -> preregistrations

Line 194: analysis -> analyses

Line 398: hull -> full

Reviewer #2: Review of Bakker et al., “We have the (1-B)! Requirements in preregistrations and IRB proposals promote formal power analyses”, submitted to PLoS ONE.

Summary

This study reports the likelihood that preregistration documents and IRB proposals incorporate a formal power analysis. It turns out that forms that request a power analysis are more likely to include one, but that this makes no difference to the proposed sample size. Some additional findings are reported regarding the amount of information provided and whether this is sufficient to reproduce the power analysis. The statistical analyses in the paper are conducted appropriately, and it is a topic of current interest given the drive to increase statistical power in empirical studies. I have some minor comments and suggestions.

Specific points

1. From hypothesis 3, the study expected to find that documents involving a power analysis would result in studies with larger sample sizes. Had this prediction been supported by the data, it would have been good evidence favouring the use of power analysis in preregistration. Do the authors feel that the converse might be true? Does the lack of an effect on sample size mean that mandating power analysis at preregistration is not really a good way to increase power, or might the null result be a type 2 error? One way to distinguish these possibilities might be to calculate a Bayes Factor score.

2. Around line 163, the paper describes how a sample of 53 studies was generated from each of two types of preregistration. But it is (somewhat ironically given the topic of the paper) unclear at this point in the manuscript where this sample size comes from, and why it is much lower than the sample size for IRB proposals.

3. It is straightforward to convert between the effect sizes in Table 2 (and the other effect sizes not included in the table). Why not convert everything to an equivalent Cohen’s d, for easier comparison across columns and to increase the number of observations?

4. This is a matter of personal taste, but the first part of the title seems superfluous and gimicky to me. I’m not even sure what it really means - why do we have the power?

5. Line 62, the authors suggest that running well powered studies will ‘decrease the need’ for researchers to engage in questionable research practices. This perhaps needs rewording, as nobody ‘needs’ to engage in this sort of thing.

6. Line 91, the authors note that funded studies have larger sample sizes than those not receiving external funding. Worth noting explicitly the obvious explanation for this, that funded studies will generally have the resources to test a larger sample.

7. Line 146, not sure what ‘next to a’ means here, please reword for clarity.

8. Line 398: hull -> full?

9. Line 412: ‘priory’ should be ‘priori’

10. Line 445: not sure what ‘effect size argumentation’ is. Do you mean the explanation of where the estimated effect size comes from?

6. PLOS authors have the option to publish the peer review history of their article (what does this mean?). If published, this will include your full peer review and any attached files.

Reviewer #1: No

Reviewer #2: Yes: Daniel H. Baker

---

## [Author Response · Author response to Decision Letter 0]

22 May 2020

(see also the cover letter)

5. Review Comments to the Author

Reviewer #1: In this preregistered study, the authors tested the effects of pre-registration type (OSF Standard Pre-Data Collection, OSF Prereg Challenge, Tilburg School of Behavioral and Social Sciences Institutional Review Board proposal) on researchers’ decisions about the rationale for the determination of sample size (e.g., whether a power analysis is conducted or not) and on the intended sample size. In line with the authors’ hypotheses, the results showed that the use of power analysis was more frequent in studies that were pre-registered according to the guidelines of OSF Prereg Challenge (PCR) form and of Tilburg School of Behavioral and Social Sciences Institutional Review Board (IRB) proposal than in studies that were pre-registered according to the guidelines of OSF Pre-Data Collection (SPR) form. Inconsistently with the authors’ hypotheses, the intended sample size was not affected by pre-registration type. The results also showed that power analyses are often characterized by technical mistakes and/or by incomplete descriptions.

Overall the manuscript is clear and well-written, and it focuses on a highly debated issue such as the reliability of psychological research. I agree with the authors’ concerns and recommendations about the need for well-conducted power analyses to increase the reliability of psychological research. Nevertheless, as regards this specific study, I have some important concerns that preclude my recommendation for publication in Plos One.

MAJOR POINTS

1) The possible impact of the results. It is unclear to me how the current study can contribute to the extant literature on research methods in psychology. The first result, which is in line with the authors’ expectation, is that power analysis is more likely to be performed when researchers are required to do it, but in my view, this just confirms the obvious. The second (and this time unexpected) finding is that pre-registration type does not appear to affect the intended sample size. The meaning of this finding remains obscure, but I suspect that it can be partly explained by the fact that, in SPRs, power analyses might have been conducted but not reported (see Major Point 3). The third result is that power analyses are often conducted in inappropriate ways (e.g., estimating effect size on a single previous study), and/or they are incorrectly reported. In my view, this is the most interesting and potentially useful outcome of this study, as it shows that simply recommending (or forcing) researchers to conduct a power analysis does not guarantee the reliability of psychological research. I believe that this manuscript would have been more interesting if it focused on this latter point.

Response: We thank the reviewer for these comments. We agree that we only know whether researchers reported a power analysis and not whether they indeed conducted a power analysis (they might have done so, but not reported it in the preregistration). This difference was sometimes unclear in our manuscript, and we made some changes in the text to make clear that it is about reporting a power analysis (e.g., line 429, 481, 486). 

We also agree that result 3 is an important finding. We restructured and rewrote the discussion of our manuscript to make this point more prominent (discussion start in row 505, directly after the discurssion of the preregistered hypotheses). 

2) The rationale of the pre-registered hypotheses and interpretation of the results. The authors suggest that both the IRB proposal and the PCR form require the participants to conduct a formal power analysis. However, as far as I could see from the OSF website, the PCR form (https://osf.io/dn3c4/) simply asks the researchers to clearly state the rationale for sample size determination. It appears that arbitrary constraints such as time and money are considered to be adequate justifications. Therefore, the PCR does not put that emphasis on power analysis, as it is instead suggested by the authors. On the contrary, power analysis probably plays a crucial role in the approval of IRB proposals. Therefore, I do not understand why the authors included PCRs and IRB proposals in the same group. These a priori objections are supported by the results, which show that the relative frequency of power analysis was much higher for IRB proposals (79%) than for PCRs (54%). On page 11, the authors affirm: “Our first hypothesis was supported; of the 207 PCRs and IRB proposals 150 (72%) made a power-based sample size decision, whereas 24 (45%) of the 53 SPRs made a power-based sample size decision …”. I believe that this is a highly misleading description of the results, due to the inappropriate averaging of the results for IRB proposals and PCRs. The percentage of PCRs in which a power analysis was performed (54%) was much more similar to that of SPRs (45%) than to that of IRB proposals (79%). I think that these objections cast severe doubts both on the rationale of the pre-registered hypotheses and on the interpretation of the results.

Response: We thank the reviewer again for these comments. We want to make clear that both the guidelines for the PCR as the IRB proposal form do not require a power analysis. Still, both require a sample size rationale and suggest to do a power analysis (PCR: “Sample size rationale. This could include a power analysis or an arbitrary constraint such as time, money, or personnel.”) or state that this is preferably done with a program like g Power (IRB proposals: “Calculation and argumentation of effect and sample size (per experiment in case of a research line) (please try to use appropriate software (e.g. G*Power) for the calculation of sample sizes).”). We already made this clear in our manuscript, but agree that, for example, the title might have suggested otherwise. We, therefore, made some changes in the text to clarify this and changed the title to Recommendations in preregistrations and internal review board proposals promote formal power analyses but do not increase sample size (see also Minor point 1).

Because both PCR and IRB require a sample size rationale and mention power analysis, we still think that we can combine the two to test our hypothesis. Furthermore, our hypotheses are preregistered and can of course not be changed based on the results that we found. Of course, differences between PCR and IRB exist, and these differences might explain the differences between the number of reported power analyses in the PCR and IRB proposals. One of these differences, which is already discussed in our manuscript, is that the review form of the IRB proposals explicitly asks about the power analysis where the review instructions of the PCR do not. Another explanation might be that researchers who were more confident about their sample size rationale shared their proposals with us and thereby biasing the results. We extended the discussion and made it more clear that this might be an alternative explanation of the results of our first hypothesis (see line 549 and further). 

3) Interpretation of the results. I believe that there is some confusion between reporting and conducting a power analysis. For the PCRs and the IRB proposals, it seems reasonable to assume that if a power analysis was not reported then it had not been conducted. However, this assumption seems to be much less reasonable for the SPRs. Indeed, the SPR form does not require the researchers to specify the rationale for the determination of sample size, which leaves open the possibility that, at least in some cases, a power analysis was actually conducted but it was simply not reported. This appears to invalidate the rationale of the pre-registered hypotheses

Response: As stated above (point 1), we made the distinction between reporting and conducting a power analysis more evident in the manuscript. We also added to our discussion of the limitations of the results of our first hypothesis, that a power analysis might be conducted but not reported in the SPRs, as suggested by the reviewer (see line 595 and further). Although we expect that the same argument applies for the SPR: when a researcher who is writing a preregistration does a power analysis, he/she will probably report it in the preregistration. We, therefore, do not agree with the reviewer that this would invalidate the rationale of our pre-registered hypotheses.

MINOR POINTS

1) Title: I believe that the use of acronyms should be avoided in the title.

Response: We changed the title to: Recommendations in preregistrations and internal review board proposals promote formal power analyses but do not increase sample size 

2) Table 1, “Sample size based on…”: the sum of the percentages is much higher than 100%. Am I missing something?

Response: As stated in the first paragraph of the scoring protocol, multiple categories could be selected (e.g., both a power analysis and some practical constraint could be mentioned). 

3) Typos:

Line 127: preregistration -> preregistrations

Line 194: analysis -> analyses

Line 398: hull -> full

Response: Thank you for your close reading. We corrected the typos.

Reviewer #2: Review of Bakker et al., “We have the (1-B)! Requirements in preregistrations and IRB proposals promote formal power analyses”, submitted to PLoS ONE.

Summary

This study reports the likelihood that preregistration documents and IRB proposals incorporate a formal power analysis. It turns out that forms that request a power analysis are more likely to include one, but that this makes no difference to the proposed sample size. Some additional findings are reported regarding the amount of information provided and whether this is sufficient to reproduce the power analysis. The statistical analyses in the paper are conducted appropriately, and it is a topic of current interest given the drive to increase statistical power in empirical studies. I have some minor comments and suggestions.

Specific points

1. From hypothesis 3, the study expected to find that documents involving a power analysis would result in studies with larger sample sizes. Had this prediction been supported by the data, it would have been good evidence favouring the use of power analysis in preregistration. Do the authors feel that the converse might be true? Does the lack of an effect on sample size mean that mandating power analysis at preregistration is not really a good way to increase power, or might the null result be a type 2 error? One way to distinguish these possibilities might be to calculate a Bayes Factor score.

Response: Thank you for your helpful comments. As stated in the manuscript there might be other reasons than mandating a power analysis for not finding an effect. One is that often the information about the sample size is missing in the SPRs. In a follow-up study it would be good to look also at the final sample size as reported in the paper and not only at the intended sample size. We agree, however, with the reviewer that it is informative to calculate the Bayes factor to have an estimate for the evidence for the null hypothesis. We found substantial evidence for the null hypothesis of no effect (BF01 = 5.280) and reported this test in our manuscript.

2. Around line 163, the paper describes how a sample of 53 studies was generated from each of two types of preregistration. But it is (somewhat ironically given the topic of the paper) unclear at this point in the manuscript where this sample size comes from, and why it is much lower than the sample size for IRB proposals.

Response: Thank you for your comments. In the part about the statistical analyses in our original submitted manuscript, we already gave a more extensive description of our power analysis (currently row 277 and further). We extended this part somewhat more, to make more clear where the different numbers come from.

3. It is straightforward to convert between the effect sizes in Table 2 (and the other effect sizes not included in the table). Why not convert everything to an equivalent Cohen’s d, for easier comparison across columns and to increase the number of observations?

Response: Thanks again for your valuable comment. We have transformed the effect sizes to Cohen’s d (if possible) and added the mean, median, and range to Table 2. We also tested whether the three registration types differed in effect size, but we did not find a significant difference (results are presented in the text). 

4. This is a matter of personal taste, but the first part of the title seems superfluous and gimicky to me. I’m not even sure what it really means - why do we have the power?

Response: We changed the title to: Recommendations in preregistrations and internal review board proposals promote formal power analyses but do not increase sample size

5. Line 62, the authors suggest that running well powered studies will ‘decrease the need’ for researchers to engage in questionable research practices. This perhaps needs rewording, as nobody ‘needs’ to engage in this sort of thing.

Response: We agree that researchers don’t need to engage in QRPs. We changed the wording and made it more clear why researchers will be less inclined to use QPRs in well-powered studies. 

6. Line 91, the authors note that funded studies have larger sample sizes than those not receiving external funding. Worth noting explicitly the obvious explanation for this, that funded studies will generally have the resources to test a larger sample.

Response: We added a sentence to the manuscript to make this explanation explicit. 

7. Line 146, not sure what ‘next to a’ means here, please reword for clarity.

Response: We reworded this sentence slightly for clarity.

8. Line 398: hull -> full?

Response: Thank you for your close reading. We changed this typo. 

9. Line 412: ‘priory’ should be ‘priori’

Response: We changed this typo.

10. Line 445: not sure what ‘effect size argumentation’ is. Do you mean the explanation of where the estimated effect size comes from?

Response: Yes, we meant that, but agree that it isn’t clear and changed the wording accordingly.

---

## [Decision Letter · Decision Letter 1]

30 Jun 2020

Recommendations in pre-registrations and internal review board proposals promote formal power analyses but do not increase sample size

PONE-D-19-20027R1

Dear Dr. Bakker,

We’re pleased to inform you that your manuscript has been judged scientifically suitable for publication and will be formally accepted for publication once it meets all outstanding technical requirements.

Kind regards,

Mariagrazia Benassi

Academic Editor

PLOS ONE

Additional Editor Comments (optional):

Reviewers' comments:

Reviewer's Responses to Questions

**Comments to the Author**

1. If the authors have adequately addressed your comments raised in a previous round of review and you feel that this manuscript is now acceptable for publication, you may indicate that here to bypass the “Comments to the Author” section, enter your conflict of interest statement in the “Confidential to Editor” section, and submit your "Accept" recommendation.

Reviewer #1: All comments have been addressed

Reviewer #2: All comments have been addressed

2. Is the manuscript technically sound, and do the data support the conclusions?

Reviewer #1: (No Response)

Reviewer #2: (No Response)

3. Has the statistical analysis been performed appropriately and rigorously? 

Reviewer #1: (No Response)

Reviewer #2: (No Response)

4. Have the authors made all data underlying the findings in their manuscript fully available?

Reviewer #1: (No Response)

Reviewer #2: (No Response)

5. Is the manuscript presented in an intelligible fashion and written in standard English?

Reviewer #1: (No Response)

Reviewer #2: (No Response)

6. Review Comments to the Author

Reviewer #1: (No Response)

Reviewer #2: (No Response)

7. PLOS authors have the option to publish the peer review history of their article (what does this mean?). If published, this will include your full peer review and any attached files.

Reviewer #1: No

Reviewer #2: No

---

## [Editor Report · Acceptance letter]

20 Jul 2020

PONE-D-19-20027R1 

Recommendations in pre-registrations and internal review board proposals promote formal power analyses but do not increase sample size 

Dear Dr. Bakker:

I'm pleased to inform you that your manuscript has been deemed suitable for publication in PLOS ONE. Congratulations! Your manuscript is now with our production department. 

Kind regards, 

on behalf of

Dr. Mariagrazia Benassi 

Academic Editor

PLOS ONE